# Factors Affecting COVID-19 Vaccination Intentions among Patients Attending a Large HIV Treatment Clinic in Trinidad Using Constructs of the Health Belief Model

**DOI:** 10.3390/vaccines11010004

**Published:** 2022-12-20

**Authors:** Nyla Lyons, Brendon Bhagwandeen, Jeffrey Edwards

**Affiliations:** 1Medical Research Foundation of Trinidad and Tobago, 7 Queen’s Part E, Port-of-Spain, Trinidad and Tobago; 2School of Mathematical and Computer Sciences, Heriot Watt University Malaysia, 1 Jalan Venna P5/2, Precinct 5, Putrajaya 62200, Malaysia; 3Department of Para-Clinical Sciences, Faculty of Medical Sciences, University of the West Indies, St. Augustine, Trinidad and Tobago

**Keywords:** COVID-19, persons living with HIV, Trinidad, vaccination intention

## Abstract

Persons living with HIV are particularly vulnerable to COVID-19 and understanding the factors influencing their decision to take the COVID-19 vaccine are crucial. Using the Health Belief Model (HBM), our study examined the role of psychological factors in predicting vaccine intention in patients with HIV. The underlying concept of the HBM is that behaviour is determined by personal beliefs about a disease, and access to strategies to decrease its occurrence. A cross-sectional survey using a structured questionnaire was conducted between August and September 2021 at an HIV clinic in Trinidad. Data on the HBM constructs, namely patient’s beliefs about the perceived severity and susceptibility to COVID-19, their perceived benefits of taking the vaccine, and external cues to action, i.e., factors that may motivate them to take the vaccine, were collected. Univariate and multivariable logistic regression analyses were used to examine associations and whether the HBM components were predictors of vaccination intention. In this study, 59.9% of patients indicated their intentions to take the vaccine. Females (OR 0.49, 95% CI 0.30–0.81) were less inclined to take the COVID-19 vaccine compared to males, while Indo-Trinidadian patients with HIV (OR 4.40, 95% CI 1.26–15.3) were more inclined to take the vaccine compared to Afro-Trinidadians. Health beliefs such as having confidence in the vaccine (*p* = 0.001) and believing in its perceived benefits (*p* = 0.001) were significant predictors of vaccination intention. Patients who were confident about the vaccine were six times more likely to take the vaccine (OR 6.45, 95% CI 2.13–19.5) than persons who were not confident in it. Having adequate information about the vaccine or the knowledge of others who received the vaccine (OR 1.48, 95% CI 1.03–2.11) were significant cues to action influencing their decision. Guided by the HBM, understanding patient’s health beliefs is important in the design of tailored interventions to improve vaccine outcomes. The HBM may also be useful in the design of approaches to increase the uptake of critical HIV prevention, and treatment services.

## 1. Introduction

People living with HIV (PLHIV) are at increased risk of severe COVID-19 disease and should be prioritized for early vaccinations [1]. HIV infection is also shown to be an independent risk factor for patients with COVID-19 infections, and independently associated with acute presentations and higher mortality during hospitalizations [2,3,4]. Consequently, every effort should be made to encourage vaccine uptake especially among persons who are moderately or severely immunocompromised such as PLHIV, and to understand the factors affecting their decisions to take the vaccine.

The Health Belief Model (HBM) is a theoretical model of behavioural change that has been used extensively to understand health prevention behaviours, including vaccination intentions among persons across various settings against papillomavirus (HPV), influenza, H1N1, and SARS-CoV-2 virus [5,6,7,8].

The HBM comprises five main constructs, namely perceived susceptibility, perceived severity, perceived benefits, perceived barriers, and cues to action. Perceived susceptibility refers to beliefs regarding vulnerability to infection, while perceived severity refers to beliefs regarding the negative effects of contracting the infection. In relation to the COVID-19 vaccination, perceived benefits are defined as an individual’s beliefs about being vaccinated, and perceived barriers are the beliefs about the obstacles to performing a recommended health action. Cues to action include information, people, and events that guide an individual’s decision on whether to be vaccinated or not [5,9].

Prior to the COVID-19 pandemic, formative research using focus groups was used to examine the associations between the HBM constructs, namely perceived threat of the disease, barriers, and cues to action, in determining flu or H1N1 vaccine intention [10]. Of the 74 persons interviewed in the study, intentions to get vaccinated were highest among persons who perceived the flu as a threat. Perceived barriers relating to vaccine access also influenced whether persons would get vaccinated. In the context of the COVID-19 pandemic, the use of the HBM has been demonstrated across several studies. In a cross-sectional study conducted on a sample of persons in France, 30.5% indicated their intentions to get vaccinated against COVID-19 during the initial vaccine implementation, while 31.1% indicated they were unsure about the vaccine [11]. In the study, the HBM constructs, namely perceptions of the risk of contracting COVID-19, and individual and collective beliefs about the benefits of the vaccine were positively associated with intentions to take it. In February 2021, a national cross-sectional survey was conducted among the adult population in Bangladesh to assess the prevalence of the COVID-19 vaccine hesitancy and its associated factors [12]. Using both online and face-to-face interviews, the results demonstrated that of the 497 respondents, the prevalence of hesitancy was 46.2%. Constructs of the HBM, namely the perceived benefits of the COVID-19 vaccines, were negatively associated with vaccine hesitancy, while perceived barriers were positively associated with vaccine hesitancy. In another cross-sectional study conducted between April and August 2021, the HBM constructs were used to examine factors influencing vaccine hesitancy among diabetes patients in China [13]. The study reported 56.4% hesitancy among 483 patients who participated. The HBM model constructs, namely the perceived risks of the COVID-19 disease, concerns regarding the safety of the vaccine, and disagreement with the cues to action (such as the doctor’s view that vaccines reduce the risk of COVID-19 infection) were independently associated with vaccine hesitancy. Further, the constructs of the HBM model were integrated with the theory of planned behaviour in a cross-sectional study examining the predictors of vaccine intentions among cancer patients [7].

The results of these studies show that HBM is useful in predicting COVID-19 vaccine hesitancy. Most studies consistently cited factors such as gender, education, age, occupation, income, employment, marital status, race, and ethnicity as having modifying effects on vaccine hesitancy and/or intentions.

The twin island republic of Trinidad and Tobago, with an estimated population of 1.39 million persons in mid-2021 [14], reported its first confirmed case of COVID-19 in March 2020. By the end of May 2022, 161,000 positive cases were reported and a total of 3909 deaths [15]. In February 2021, COVID-19 vaccinations were made available to the public. Several cross-sectional studies were implemented and examined public willingness to take the COVID-19 vaccine. In May 2020, a regional study conducted by the Johns Hopkins Centre for Communication Programs, which included participants from Trinidad and Tobago, found that 59% of persons were not willing to accept the COVID-19 vaccine [16]. Later in December 2020, an exploratory online survey examining public willingness to receive the COVID-19 vaccine found that 62.8% of the 643 participants indicated that they would take the vaccine once it was available. The study found lower levels of health literacy were associated with higher levels of misinformation about the vaccine. Study participants reported that information from health workers (32.5%) and the Ministry of Health (23.6%) were the two most trusted sources of information about the COVID-19 vaccines [17]. Once the COVID-19 vaccines became publicly available, a survey conducted by Market Facts and Opinions (MFO) examined public perceptions of the vaccines and indicated that 65% (of a random sample of 973 participants) were not willing to take the COVID-19 vaccines [18]. Later in August 2021, a regional study conducted by UNICEF reported that among participants from Trinidad and Tobago, 65% of the survey sample reported they already took the COVID-19 vaccination. Among the unvaccinated, 37% felt that the vaccines were unsafe and were unsure of their long-term side effects [19]. The factors affecting the uptake of COVID-19 vaccinations among persons living with HIV have not been studied in the local context of Trinidad. Internationally, the available studies examining vaccination uptake among PLHIV have cited barriers such as accessibility, affordability, and mistrust as limiting vaccination uptake and reducing a patient’s confidence in the benefits of the vaccinations [3,20,21]. Less has been published on the contribution of psychological factors on COVID-19 vaccine intentions among PLHIV. Using the constructs of the HBM, this study examined the role of beliefs and psychological factors in predicting vaccination intentions among patients with HIV attending a large treatment clinic in Trinidad; exploring significant HBM constructs may be crucial in the design of tailored interventions to improve decisions to get vaccinated against COVID-19 and reduce the consequences for disease mortality associated with HIV infection.

## 2. Materials and Methods

### 2.1. Study Design and Participants

This cross-sectional study was carried out from August to September 2021 among persons living with HIV in Trinidad and attending the Medical Research Foundation of Trinidad and Tobago (MRFTT) for HIV treatment and care, by face-to-face interviews using a structured questionnaire. The MRFTT is the largest HIV Treatment and Care Centre in Trinidad and Tobago, where daily clinics are held via appointments and walk-in visits.

Due to the COVID-19 pandemic, the MRFTT implemented measures to reduce crowdedness at the clinic and mitigate the potential spread of the virus among patients. These included telehealth, ART delivery, and 6-month medication prescriptions. As a result, scheduled patient appointments fell by 75%, to approximately 400 patients per month. Therefore, a convenience sample was collected based on the total number of patients expected to attend the MRFTT for HIV treatment and care over the two-month period during which this study was conducted. A sampling quota was used to reduce some of the potential sampling bias. This quota was based on participant gender (male and female, 1:1 ratio). Assuming approximately 800 patients would attend the clinic over the two-month period; using a margin of error of 5%, a confidence level of 95%, and a 50% vaccination intention rate, this gave a minimum sample size of 260 participants. The final sample comprised *n* = 272 patients who met the eligibility criteria and were willing to participate.

The eligibility criteria included persons in Trinidad and Tobago living with HIV, aged 18 years and older, and attending the MRFTT for HIV treatment and care. These persons received information about the study protocol and procedures and participated after their written consents were provided. To confirm the quality of the interviews, nurses at the MRFTT were trained on data collection and the inclusion criteria procedure. Participant confidentiality was maintained throughout the survey, and persons were also informed that they could stop completing the survey at any time. All physical copies of the completed questionnaires were secured at the MRFTT, and responses were transferred to an Excel spreadsheet stored on a password protected computer, with only members of the research team having access to this data file.

### 2.2. Ethical Approval

The study protocol, data collection instruments and informed consent form were approved by the Campus Research Ethics Committee of the University of the West Indies, St. Augustine, Trinidad, approval number CREC-SA.1063/06/2021.

### 2.3. Study Questionnaire and Data Collection

A structured questionnaire was administered. This included socio-demographic questions, health-related questions, and questions based on the HBM.

The questionnaire consisted of the following sections: (1) socio-demographic factors which included age, gender, ethnicity, marital status, education level, employment status, and sexual orientation (1 = non heterosexual adults (i.e., gay and lesbian); 2 = heterosexual adults); health-related questions which included (2) presence of chronic diseases (such as cancer, cardiovascular disease, diabetes, respiratory diseases), (3) self-rated health, (4) previous or current exposure to COVID-19; (5) outcome/dependent variable which was COVID-19 vaccination intention; and (6) psychological constructs of the HBM which included perceived susceptibility of contracting COVID-19, perceived severity of COVID-19, perceived benefits about the COVID-19 vaccine, perceived barriers to COVID-19 vaccination, and cues to action. The questionnaire took between 10 and 15 min to complete. Table 1 provides further details of the variables and measurements in this study.

Perceived susceptibility of contracting COVID-19, perceived severity of COVID-19, perceived benefits of the COVID-19 vaccine, perceived barriers to COVID-19 vaccination, and cues to action were scored on a 4-point Likert scale, with a score of 1 indicating strong disagreement and a score of 4 indicating strong agreement. Item 4: “The medications I am currently taking can protect me from contracting COVID-19 infections”, of the perceived susceptibility of contracting COVID-19 construct, was reverse coded to maintain internal consistency. A score was then created to measure each construct using the sum of scores of the responses from the Likert scales.

A standard median split was performed on the dependent variable “COVID-19 vaccination intention” (previously measured by a three-scale item) transforming it into a dichotomous variable. Scores that fell below the median were put into a “negative” group (0 = do not intend to take the COVID-19 vaccine) and scores that were above the median were put into a “positive” group (1 = intend to take the COVID-19 vaccine). Sensitivity analysis was conducted and showed no differences in conclusions when using the continuous and dichotomous vaccination intention outcome variables. Therefore, the dichotomous outcome variable was used in the final analysis for this study to facilitate comparison of COVID-19 vaccination intentions with other studies and meaningful interpretations in terms of odds ratios.

### 2.4. Other Factors

Viral suppression, and the length of time since HIV diagnosis and initiation on antiretroviral therapy (ART) were retrieved from patient records. Viral suppression, as a measure of treatment success, was indicated by the results of patient’s last viral load (VL) test within 12 months or less from the study implementation date. Patients with a recorded viral load of 1000 copies per ml or less (VL < 1000 copies/mL) were considered “virally suppressed” in the context of this study.

### 2.5. Reliability of the Questionnaire

A pilot study was conducted to validate the questionnaire. Medical doctors and medical research experts at the MRFTT reviewed the questionnaire design, content, words, comprehension, and ease of completion.

The Cronbach α coefficients for the HBM components showed low to high internal consistency. The coefficient for perceived susceptibility to the COVID-19 disease was 0.55; cues to action scale was 0.58; perceived fears of COVID19 vaccination was 0.86; perceived barriers about COVID-19 vaccination was 0.88; and perceived severity of COVID-19 was 0.78. Although perceived susceptibility to the COVID-19 disease and cues to action components had relatively low coefficients, medical professionals at the MRFTT believed analyses of these perceptions were important and relevant in the context of the study and the COVID-19 pandemic. Therefore, these components were retained in the final questionnaire.

### 2.6. Statistical Analysis

The Statistical Package for the Social Sciences (SPSS) v22.0 [22] was used for statistical analysis. Descriptive statistics (mean ± standard deviation for continuous variables and n (%) for categorical variables) were used to characterize baseline distributions of study variables. Baseline differences were compared using independent samples t-tests for continuous variables and Pearson χ^2^ test for categorical variables. Descriptive statistics were also used to report COVID-19 vaccination intention. Univariate analyses examined the relationship between vaccine intention and baseline characteristics. Explanatory logistic regression was used to examine unadjusted and adjusted odds ratios (OR) (with 95% confidence intervals (CI)) between COVID-19 vaccination intention and psychological research constructs, controlling for potential sociodemographic confounders. A priori possible predictors of COVID-19 vaccine intention with *p* < 0.05 in univariate analyses were considered in the multivariable model. Analyses included only non-missing data. *p* < 0.05 was considered statistically significant.

## 3. Results

### 3.1. Characteristics of Participants Living with HIV

A total of 272 patients attending an HIV treatment clinic in Trinidad and Tobago completed the survey, of whom 135 (49.6%) were males and 137 (50.4%) were females. The ethnicity distribution showed 161 (59.2%) participants of African descent, 25 (9.2%), of East Indian descent, and 84 (30.9%) were either mixed or belonged to other ethnicities. Twenty-eight participants (10.3%) were married, 179 (65.8%) were single, while the remaining 65 (23.9%) were in a common law relationship. Forty-one participants (15.1%) received schooling up to the primary school level, 172 (63.2%) received up to secondary schooling, while 58 (21.3%) completed university studies. Regarding employment, 137 participants (50.4%) were employed, while 107 (39.3%) were unemployed. Of the participants in the study, 192 (70.6%) were heterosexual (refer to Table 2).

Most participants (*n* = 249, 91.5%) did not have any prior experiences with contracting the COVID-19 virus. However, 70 (25.7%) had either family members or friends who contracted COVID-19. Seventy-three participants (26.8%) had existing chronic diseases; however, based on self-reports, 217 participants (79.8%) believed that they were in good health (refer to Table 2).

One hundred and twenty-nine persons (47.4%) indicated that they were confident in the COVID-19 vaccine, while 139 (51.1%) were not confident. In the overall sample, 163 persons (59.9%) had intentions to take the COVID-19 vaccine (refer to Table 2).

Of the total study sample, 229 patients (84%) were virally suppressed, i.e., had a recorded viral load (VL) test result of 1000 copies or less within 12 months or less of the study implementation date. The mean age of the patients enrolled in our study was 41 ± 13.0 years, and patients were initiated on antiretroviral therapy (ART) for an average 8.73 ± 6.02 years since the time of initial diagnosis (refer to Table 2).

### 3.2. Differences in Intention to Receive the COVID-19 Vaccine and Baseline Characteristics

Significant differences in COVID-19 vaccination intention were not observed when categories of marital status (*p* = 0.523), education (*p* = 0.224), employment status (*p* = 0.164), prior experience with COVID-19 (*p* = 0.824), whether family/friends contracted COVID-19 (*p* = 0.138), presence of chronic diseases (*p* = 0.560), self-rated health status (*p* = 0.919), and viral suppression (*p* = 0.955) were considered. There were also no significant differences when age (*p* = 0.304), and length of time since HIV diagnosis and initiation on ART (*p* = 0.486) were used as independent variables. However, significant differences in COVID-19 vaccination intention were present when categories of gender (*p* = 0.005), ethnicity (*p* = 0.001), sexual orientation (*p* = 0.047), and confidence in the COVID-19 vaccine (*p* < 0.001) were considered. There was a greater proportion of males (68.9%) who intended to get vaccinated against COVID-19 compared to females (52.2%). Indo-Trinidadians (88%) had the highest proportion for persons who intended to get vaccinated, followed by Afro-Trinidadians (62.5%), and persons of mixed/other ethnicities (47.6%). Non-heterosexuals (71.6%) had a higher proportion of persons intending to take the COVID-19 vaccine compared to heterosexuals (57.9%). The proportion of persons who intended to get vaccinated given that they were confident in the COVID-19 vaccine (91.3%) was greater than the proportion of persons who intended to get vaccinated but were not confident in it (31.7%) (refer to Table 3).

### 3.3. Associations between Intention to Receive the COVID-19 Vaccine and Baseline Characteristics

Gender was significantly associated with COVID-19 vaccination intention (*p* = 0.005). Females were less likely (OR 0.49, 95% CI 0.30–0.81) to take the COVID-19 vaccine relative to males. Ethnicity was also significantly associated with COVID-19 vaccination intention in this study (*p* = 0.001). Indo-Trinidadians were over four times more likely (OR 4.40, 95% CI 1.26–15.3) to take the COVID-19 vaccine, but persons of mixed ethnicity were less likely (OR 0.54, 95% CI 0.32–0.93) to take the vaccine relative to Afro-Trinidadians. Sexual orientation was another factor significantly associated with COVID-19 vaccination intention (*p* = 0.047). Non-heterosexual persons were almost twice likely (OR 1.84, 95% CI 1.00–3.36) to take the COVID-19 vaccine relative to heterosexual persons. Patient confidence in the COVID-19 vaccine was also significantly associated with vaccination intention (*p* < 0.001). Persons who were confident in the COVID-19 vaccine were considerably more likely (OR 22.8, 95% CI 11.1–46.5) to take the COVID-19 vaccine relative to persons who were not confident. There were no significant associations with marital status, education level, employment status, experience with COVID-19, family/friends contracting COVID-19, existing chronic diseases, self-rated health status, viral suppression, age or length of time since HIV diagnosis, and initiation on ART and COVID-19 vaccination intention (*p* > 0.05) (refer to Table 4).

### 3.4. The HBM Constructs and COVID-19 Vaccination Intention 

Table 5 and Table 6 provide descriptive statistics for the individual psychological constructs. The psychological constructs of the HBM were found to be significantly associated with COVID-19 vaccination intention (*p* < 0.05).

Higher scores on the perceived susceptibility of contracting COVID-19 scale were associated with COVID-19 vaccination intention (do not intend to get vaccinated: 9.74 ± 2.27, intend to get vaccinated: 10.5 ± 2.09; *p* = 0.011) (refer to Table 5).

Lower scores on the perceived severity of COVID-19 scale were associated with COVID-19 vaccination intention (do not intend to get vaccinated: 8.95 ± 2.00, intend to get vaccinated: 6.82 ± 1.88; *p* < 0.001) (refer to Table 5). Proportions of persons strongly agreeing/agreeing with the three scale items were significantly less for persons who intended to get vaccinated against COVID-19 compared to persons who did not intend to get vaccinated (*p* < 0.001) (refer to Table 6).

Higher scores on the perceived benefits of the COVID-19 vaccine scale were associated with COVID-19 vaccination intention (do not intend to get vaccinated: 4.24 ± 1.40, intend to get vaccinated: 6.01 ± 1.22; *p* < 0.001) (refer to Table 5). Proportions of persons strongly agreeing/agreeing with the two scale items were significantly greater for persons who intended to get vaccinated against COVID-19 compared to persons who did not intend to get vaccinated (*p* < 0.001) (refer to Table 6).

Lower scores on the perceived barriers to COVID-19 vaccination scale was associated with high COVID-19 vaccination intention (do not intend to get vaccinated: 16.0 ± 3.03, intend to get vaccinated: 12.6 ± 3.37; *p* < 0.001) (refer to Table 5). Proportions of persons strongly agreeing/agreeing with the five scale items were significantly less for persons who intended to get vaccinated against COVID-19 compared to persons who did not intend to get vaccinated (*p* < 0.001) (refer to Table 6).

Higher scores on the cues to action scale were associated with COVID-19 vaccination intention (do not intend to get vaccinated: 5.18 ± 1.52, intend to get vaccinated: 5.63 ± 1.25; *p* < 0.014) (refer to Table 5). Proportions of persons strongly agreeing/agreeing with the two scale items were significantly greater for persons who intended to get vaccinated against COVID-19 compared to persons who did not intend to get vaccinated (*p* < 0.05) (refer to Table 6).

### 3.5. Predictors of COVID-19 Vaccination Intention

Baseline characteristics and psychological constructs that were significantly associated with COVID-19 vaccination intention (*p* < 0.05) in our univariate analyses were included in the multivariable model to assess predictors of COVID-19 vaccination intention (refer to Table 7).

In the multivariable model, confidence in the COVID-19 vaccine was a significant predictor of COVID-19 vaccination intention (*p* = 0.001), with persons who were confident in the COVID-19 vaccine being over six times more likely to get the vaccine (OR 6.45, 95% CI 2.13–19.5) relative to persons who were not confident, while holding other factors constant. The perceived benefits of the COVID-19 vaccine (OR 1.85, 95% CI 1.27–2.70) and cues to action (OR 1.48, 95% CI 1.03–2.11) constructs were also found to be significant predictors of COVID-19 vaccination intention (*p* < 0.05) (refer to Table 7).

In the multivariable model, gender, ethnicity, sexual orientation, and the perceived susceptibility of contracting COVID-19, perceived severity of COVID-19, and perceived barriers of the COVID-19 vaccine constructs were no longer significantly associated with COVID-19 vaccination intention (*p* > 0.05) (refer to Table 7).

## 4. Discussion

In this study, the HBM constructs, namely perceptions of the COVID-19 vaccination benefits and cues to action were significant predictors of vaccination intentions among patients. These results provided further confirmation of the role of the HBM in predicting and explaining preventive health behaviours [5,6,7] such as intentions to receive the COVID-19 vaccination.

Over half (59.9%) of the patients enrolled in this study indicated their intentions to take the COVID-19 vaccination. These results support findings from some previous studies documenting moderate to high vaccination intentions among persons with HIV in Foch Hospital, Suresnes, France (71.3%) and in British Columbia, Canada (65.2%) [3,23]. This study was implemented in August through until September 2021—four months after the COVID-19 vaccines were made available in Trinidad and Tobago. This was also during the period when results from clinical trials on the efficacy of COVID-19 vaccines for PLHIV were still emerging. These factors may have contributed to just below two-thirds (59.9%) of the study sample indicating their intentions to get vaccinated against the COVID-19 virus.

Since the onset of the COVID-19 pandemic, patients enrolled in care at the MRFTT were consistently exposed to information about the COVID-19 disease. With the roll out of the national vaccination program, patients attending the MRFTT for their HIV treatment benefited from counselling and education about the COVID-19 vaccination to encourage its uptake. All clinic staff including doctors, nurses, and social workers provided vaccine-specific counselling to address patient concerns about their risk of adverse health outcomes, and reduced fears about the potential for side effects between the COVID-19 vaccination and their HIV medication regimens. While the COVID-19 vaccinations were not accessible at the MRFTT clinic at the time, staff routinely addressed patients’ concerns during their clinic visits which may have accounted for 59.9% of patients enrolled in our study reporting their intentions to take the COVID-19 vaccination.

Among the sociodemographic factors in the univariate analysis, COVID-19 vaccination intention was lower among female patients (OR 0.49, 95% CI 0.30–0.81) relative to males. Vaccination intentions were more likely among patients of East Indian descent (OR 4.40, 95% CI 1.26–15.3) relative to patients of African descent, and those identifying as non-heterosexual, i.e., gay, and lesbian adults (OR 1.84, 95% CI 1.00–3.36), relative to heterosexual persons. These results, like previous studies, showed that demographic factors such as sex and ethnicity are consistently associated with vaccination intentions. A multinational systematic review demonstrated female sex, younger age, lower income, educational level, and membership in an ethnic minority group to be consistently associated with no COVID-19 vaccine intent [9]. Studies on COVID-19 and persons identifying as members of lesbian, gay, bisexual, and transgender (LGBT) populations are limited. Data from the National Immunization Survey Adult COVID Module (NIS-ACM) were analysed to assess COVID-19 self-reported vaccination coverage and confidence in COVID-19 vaccines among LGBT adults aged ≥18 years. The findings from the study showed that COVID-19 vaccination was higher among gays and lesbian adults compared to heterosexual adults [24]. This difference may be due to non-heterosexual male patients having better health-seeking behaviour than heterosexual patients in the current study context. Therefore, addressing female patients would be important to enhance COVID-19 vaccine acceptance. In this study, there were no significant associations between the presence of chronic diseases and intentions to take the COVID-19 vaccinations. This finding was not consistent with previous studies, which found that patients with underlying comorbid disease conditions were more likely to have intentions to the take the COVID-19 vaccination than those without underlying conditions [25,26]. The difference might have been due to high levels of viral suppression among patients in our study, many of whom indicated they were in good health; therefore, having an existing health condition did not affect how they felt about receiving a vaccination at the time.

The results in this study also showed that patients who were more confident in the COVID-19 vaccines were six times more likely (OR 6.45, 95% CI 2.13–19.5) to get vaccinated against COVID-19. During the progression of the COVID-19 pandemic there was early evidence that a storm of misinformation was developing, largely from social and alternative media sources; these sources, coupled with the speed of vaccine development, were eroding public confidence in COVID-19 containment measures, and trust in the governments and health care systems [27,28]. While this study did not examine the influence of sources of COVID-19 information such as social media, and/or the trust in the public healthcare system, the results suggested that patients enrolled in care at the MRFTT tended to have more confidence in the information provided by healthcare providers regarding the COVID-19 vaccinations.

The theoretical framework of the HBM was used in this study to examine the role of psychological factors as predictors of COVID-19 vaccination intentions. According to the HBM, a person’s belief and attitudes influence their psychological behaviours and/or actions, and therefore, a change in beliefs and attitudes might result in a moderate change in health preventive behaviours. The HBM has been extensively applied in several cross-sectional studies to understand preventive health behaviours [5,6,7]. The Cronbach α coefficients for the HBM components showed low to high internal consistency. All five HBM constructs, namely the perceived susceptibility to, and severity of, COVID-19, perceived benefits and barriers to COVID-19 vaccination, and cues to action were significantly associated with COVID-19 vaccination intentions among the patients in our sample. These findings were also consistent with the health behaviour literature that persons with lower perceived severity about the COVID-19 disease, those reporting higher perceived benefits and lower perceived barriers toward vaccination were associated with intentions to take the vaccinations [29,30,31].

The multivariable analyses showed that the HBM construct, namely, the perceived benefit of COVID-19 vaccination (*p* = 0.001) was the strongest predictor of vaccination intentions, followed by cues to action (*p* = 0.029), i.e., agreement to take the vaccine when provided with external cues to action. These results indicated that patients with higher perceived benefits about the COVID-19 vaccines were almost two times as likely to get vaccinated against COVID-19 (OR 1.85, 95% CI 1.27–2.70). For the HBM construct, i.e., cues to action, the results suggest that having adequate information about the vaccines and/or the knowledge of others who have received the vaccine (OR 1.48, 95% CI 1.03–2.11) increase the likelihood of vaccination intentions in patients in this study.

Overall, the significance of the HBM constructs in predicting COVID-19 vaccination intentions among patients with HIV, further reinforced the use of this model to examine psychological factors influencing COVID-19 vaccine behaviour (including uptake of vaccinations) across various settings and across populations [29,30,31,32]. The findings from some of these studies have been mixed. One study conducted in the early phase of the COVID-19 pandemic in China showed that the HBM component “perceived susceptibility” was the strongest predictor of behaviour change [31]. During the COVID-19 pandemic, there was an increased concern among patients with autoimmune disease being at a higher risk or more prone to getting COVID-19 infection. At the same time, there was concern of them having a higher risk of severe forms of COVID-19 infection [29] and these risks were likely to affect vaccination intentions. The spread of misinformation surrounding the pandemic also posed a serious concern for COVID-19 vaccine intentions and perceptions of severity of the COVID-19 disease.

Given the overall significance of patient beliefs (as measured by the HBM) in this study, it may be crucial to target the HBM components, i.e., a patient’s confidence in perceived benefits of the COVID-19 vaccines, and cues to action as the as these played an important role in predicting vaccination intentions. The role of the healthcare providers at the facility level was, therefore, critical in increasing education and information.

Two months after the implementation of this study, COVID-19 vaccinations were available for patients attending the MRFTT clinic. The increased availability and accessibility of COVID-19 vaccinations, integrated with the delivery of HIV services for patients with HIV, was a critical component to increasing acceptance and uptake of COVID-19 vaccinations. Increased availability and accessibility, combined with targeted strategies at the facility/clinic level to encourage patients to share their positive thoughts and experiences regarding the COVID-19 vaccination, targeting women and persons from various ethnic groups, and reducing misinformation about the vaccine would bolster COVID-19 vaccine uptake efforts.

### Study Limitations

This study was not without limitations. The study was conducted during the period when COVID-19 vaccines were first made widely available, thus corresponding with heightened public misconceptions and concerns about the safety and efficacy of their use. Information regarding COVID-19 vaccinations and HIV was still evolving during the time of the implementation, contributing to uncertainty of vaccine safety among this subpopulation. Furthermore, additional unmeasured confounding factors could have influenced vaccine intention on a personal level, such as sources of information, including social media, the reliability and trust in health professionals, and facility and societal level factors such as stigma and trust in the health system. This was also during the period of COVID-19 restrictions, when several measures were implemented to reduce the potential risk of the spread of COVID-19 infections by patients and staff. Risk reduction interventions such as the use of telehealth, community ART delivery, and 6-month ART dispensing substantially reduced the number of in-person visits and, therefore, the study sample may not be representative of the entire clinic population. The sample used was a convenient sample of participants, recruited among patients attending an HIV treatment clinic, and therefore the results were limited to patients who were enrolled in the study. As it pertains to the HBM, the model does not suggest a strategy for changing health-related actions.

## 5. Conclusions

The present study reinforced the use of the HBM to predict health prevention behaviours, such as patients’ intentions to get vaccinated against COVID-19. It was important to understand the determinants of individual vaccination decisions to establish effective strategies to increase the uptake of COVID-19 vaccinations among PLHIV and other high-risk populations. The study underscored the role of psychological factors as the strongest predictors of vaccination intentions among a sample of patients with HIV. Specifically, it might be crucial to tailor interventions to increase a patient’s perceptions of the benefits of the vaccine, ensuring that adequate information is provided, and patients are encouraged to share their positive thoughts and experience with others. Our study also has implications for the use of the HBM to understand and predict behaviours regarding the uptake of critical prevention, care, and treatment interventions, and for the design of policies contributing to reducing HIV transmission among high-risk groups.

## Figures and Tables

**Table 1 vaccines-11-00004-t001:** Variables and measurement.

Dependent Variable	Measurement
COVID-19 Vaccination Intention	1. If you are offered the COVID-19 vaccine in the next 2 weeks, would you take it?
	2. If you are offered the COVID-19 vaccine in 3 to 6 months, would you take it?
	3. If you are offered the COVID-19 vaccine in one year, would you take it?
HBM Constructs	
Perceived susceptibility of contracting COVID-19	1. My chance of getting COVID-19 is great.
	2. I am worried about the likelihood of getting COVID 19.
	3. My present health condition makes it easier for me to contract COVID-19 infection.
	4. The medications I am currently taking can protect me from contracting COVID-19 infections.
Perceived severity of COVID-19	1. Complications from COVID-19 vaccination are serious. 2. I will be very sick if I take the COVID-19 vaccine. 3. I am afraid of taking the COVID-19 vaccine.
Perceived benefits about the COVID-19 vaccine	1. Vaccination is a good idea because I feel less worried about catching COVID-19.
	2. Vaccination decreases my chances of getting COVID-19 or its complications.
Perceived barriers to COVID-19 vaccination	1. I am worried the possible side-effects of COVID-19 vaccination would interfere with my usual activities.
	2. I am concerned about the effectiveness of the COVID-19 vaccination.
	3. I am concerned about the safety of the COVID-19 vaccination.
	4. I am concerned that the COVID-19 vaccine needs more testing before I feel safe to take it.
	5. I am concerned of the faulty/fake COVID-19 vaccine.
Cues to action	1. I will only take the COVID-19 vaccine if I was given adequate information about it.
	2. I will only take the COVID-19 vaccine if the vaccine is taken by many in the public.

**Table 2 vaccines-11-00004-t002:** Baseline characteristics of study sample (*n* = 272).

Variable	Entire Sample (*n* = 272)	
	*n*	%
Gender		
Male	135	49.6
Female	137	50.4
Ethnicity		
African	161	59.2
Indian	25	9.2
Mixed/Other	84	30.9
Non-responder	2	0.7
Marital status		
Married	28	10.3
Single	179	65.8
Common law/Other	65	23.9
Education level		
Primary school or below	41	15.1
Secondary school	172	63.2
University or above	58	21.3
Non-responder	1	0.4
Employment status		
Employed	137	50.4
Unemployed	107	39.3
Other	28	10.3
Sexual orientation		
Heterosexual	192	70.6
Non-heterosexual	68	25.0
Non-responder	12	4.4
Previous or current exposure to COVID-19 disease		
No	249	91.5
Yes	19	7.0
Non-responder	4	1.5
Family/friends contracting COVID-19		
No	202	74.3
Yes	70	25.7
Existing chronic diseases		
No	195	71.7
Yes	73	26.8
Non-responder	4	1.5
Self-rated health status		
Poor health	55	20.2
Good health	217	79.8
Confidence in the COVID-19 vaccine		
Not confident	139	51.1
Confident	129	47.4
Non-responder	4	1.5
Viral suppression		
Not suppressed	39	14.3
Suppressed	229	84.2
COVID-19 vaccination intention		
Do not intend to get vaccinated	106	39.0
Intend to get vaccinated	163	59.9
Non-responder	3	1.1
	Mean ± SD	
Age (years)	41.7 ± 13.0	
Length of time since HIV diagnosis and initiation on ART (years)	8.73 ± 6.02	

**Table 3 vaccines-11-00004-t003:** Differences in COVID-19 vaccination intention according to baseline characteristics, row %.

Variable	COVID-19 Vaccination Intention		
	Do Not Intend to Get Vaccinated	Intend to Get Vaccinated	*p*-Value
Gender			0.005
Male	42 (31.1%)	93 (68.9%)	
Female	64 (47.8%)	70 (52.2%)	
Ethnicity			0.001
Afro-Trinidadian	60 (37.5%)	100 (62.5%)	
Indo-Trinidadian	3 (12.0%)	22 (88.0%)	
Mixed/Other	43 (52.4%)	39 (47.6%)	
Marital status			0.523
Married	13 (46.4%)	15 (53.6%)	
Single	66 (37.1%)	112 (62.9%)	
Common law/Other	27 (42.9%)	36 (57.1%)	
Education level			0.224
Primary school or below	12 (30.0%)	28 (70.0)	
Secondary school	74 (43.3%)	97 (56.7%)	
University or above	20 (35.1%)	37 (64.9%)	
Employment status			0.164
Employed	46 (33.8%)	90 (66.2%)	
Unemployed	47 (44.8%)	58 (55.2%)	
Other	13 (46.4%)	15 (53.6%)	
Sexual orientation			0.047
Heterosexual	80 (42.1%)	110 (57.9%)	
Non-heterosexual	19 (28.4%)	48 (71.6%)	
Experience with COVID-19			0.824
No	97 (39.4%)	149 (60.6%)	
Yes	7 (36.8%)	12 (63.2%)	
Family/friends contracting COVID-19			0.138
No	84 (42.0%)	116 (58.0%)	
Yes	22 (31.9%)	47 (68.1%)	
Existing chronic diseases			0.560
No	74 (38.5%)	118 (61.5%)	
Yes	31 (42.5%)	42 (57.5%)	
Self-rated health status			0.919
Poor health	22 (40.0%)	33 (60.0%)	
Good health	84 (39.3%)	130 (60.7%)	
Confidence in the COVID-19 vaccine			<0.001
Not confident	95 (68.3%)	44 (31.7%)	
Confident	11 (8.7%)	116 (91.3%)	
Viral suppression			0.955
Not suppressed	15 (38.5%)	24 (61.5%)	
Suppressed	88 (38.9%)	138 (61.1%)	
Age	40.6 ± 13.2	42.3 ± 13.0	0.304
Length of time since HIV diagnosis and initiation on ART	8.99 ± 5.98	8.47 ± 6.04	0.486

**Table 4 vaccines-11-00004-t004:** Univariate analyses and unadjusted odds ratios (with 95% CI) of COVID-19 vaccination intention and baseline characteristics.

Variable	COVID-19 Vaccination Intention (Ref: Do Not Intend to Get Vaccinated)		
	Unadjusted OR	95% CI	*p*-Value
Gender			
Male	Ref	Ref	
Female	0.49	0.30–0.81	0.005
Ethnicity			
Afro-Trinidadian	Ref	Ref	
Indo-Trinidadian	4.40	1.26–15.33	0.020
Mixed/Other	0.54	0.32–0.93	0.027
Marital status			
Married	Ref	Ref	
Single	1.47	0.66–3.28	0.346
Common law/Other	1.16	0.47–2.83	0.751
Education level			
Primary school or below	Ref	Ref	
Secondary school	0.56	0.27–1.18	0.127
University or above	0.79	0.33–1.89	0.600
Employment status			
Employed	Ref	Ref	
Unemployed	0.63	0.37–1.07	0.084
Other	0.59	0.26–1.34	0.209
Sexual orientation			
Heterosexual	Ref	Ref	
Non-heterosexual	1.84	1.00–3.36	0.048
Experience with COVID-19			
No	Ref	Ref	
Yes	1.12	0.43–2.93	0.824
Family/friends contracting COVID-19			
No	Ref	Ref	
Yes	1.55	0.87–2.76	0.140
Existing chronic diseases			
No	Ref	Ref	
Yes	0.85	0.49–1.47	0.560
Self-rated health status			
Poor health	Ref	Ref	
Good health	1.03	0.56–1.89	0.919
Confidence in the COVID-19 vaccine			
Not confident	Ref	Ref	
Confident	22.8	11.1–46.5	<0.001
Viral suppression			
Not suppressed	Ref	Ref	
Suppressed	0.98	0.49–1.97	0.955
Age	1.01	0.99–1.03	0.303
Length of time since HIV diagnosis and initiation on ART	0.99	0.95–1.03	0.484

**Table 5 vaccines-11-00004-t005:** COVID-19 vaccination intention and COVID-19 vaccine psychological construct scores.

Psychological Construct	Overall Mean Score ± SD	COVID-19 Vaccination Intention		
		Do Not Intend to Get Vaccinated	Intend to Get Vaccinated	*p*-Value
Perceived susceptibility of contracting COVID-19	10.2 ± 2.18	9.74 ± 2.27	10.5 ± 2.09	0.011
Perceived severity of COVID-19	7.65 ± 2.19	8.95 ± 2.00	6.82 ± 1.88	<0.001
Perceived benefits of the COVID-19 vaccine	5.32 ± 1.56	4.24 ± 1.40	6.01 ± 1.22	<0.001
Perceived barriers of the COVID-19 vaccine	13.9 ± 3.64	16.0 ± 3.03	12.6 ± 3.37	<0.001
Cues to action	5.46 ± 1.37	5.18 ± 1.52	5.63 ± 1.25	0.014

**Table 6 vaccines-11-00004-t006:** COVID-19 vaccination intention and COVID-19 vaccine psychological constructs by item, strongly agree/agree column %.

Psychological Construct Items	COVID-19 Vaccination Intention		
	Do Not Intend to Get Vaccinated	Intend to Get Vaccinated	*p*-Value
Perceived susceptibility of contracting COVID-19			
My chance of getting COVID-19 is great	32.4%	39.9%	0.217
I am worried about the likelihood of getting COVID-19	48.1%	58.8%	0.089
My present health condition makes it easier for me to contract COVID-19	42.3%	50.3%	0.203
The medications I am currently taking can protect me from contracting COVID-19	25.0%	21.4%	0.494
Perceived severity of COVID-19			
Complications from COVID-19 vaccination are serious	85.4%	48.1%	<0.001
I will be very sick if I take the COVID-19 vaccine	60.4%	19.1%	<0.001
I am afraid of taking the COVID-19 vaccine	74.8%	34.6%	<0.001
Perceived benefits of the COVID-19 vaccine			
Vaccination is a good idea because I feel less worried about catching COVID-19	30.8%	82.5%	<0.001
Vaccination decreases my chance of getting COVID-19 or its complications	25.0%	80.1%	<0.001
Perceived barriers of the COVID-19 vaccine			
I am worried the possible side-effects of the COVID-19 vaccine would interfere with my usual activities	76.9%	40.4%	<0.001
I am concerned about the effectiveness of the COVID-19 vaccine	82.7%	51.9%	<0.001
I am concerned about the safety of the COVID-19 vaccine	85.6%	58.4%	<0.001
I am concerned that the COVID-19 vaccine needs more testing before I feel safe to take it	89.4%	46.0%	<0.001
I am concerned of the faulty/fake COVID-19 vaccine	83.7%	49.4%	<0.001
Cues to action			
I will only take the COVID-19 vaccine if I was given adequate information about it	69.2%	86.3%	0.001
I will only take the COVID-19 vaccine if the vaccine is taken by many in the public	42.7%	55.9%	0.037

**Table 7 vaccines-11-00004-t007:** Adjusted odds ratios (with 95% CI) of COVID-19 vaccination intention with COVID-19 vaccine psychological constructs and baseline characteristics.

Variable	COVID-19 Vaccination Intention (Ref: Do Not Intend to Get Vaccinated)		
	Adjusted OR	95% CI	*p*-Value
Gender			
Male	Ref	Ref	
Female	1.46	0.60–3.54	0.400
Ethnicity			
Afro-Trinidadian	Ref	Ref	
Indo-Trinidadian	2.51	0.55–11.38	0.234
Mixed	0.42	0.17–1.04	0.061
Sexual orientation			
Heterosexual	Ref	Ref	
Non-heterosexual	2.32	0.78–6.91	0.130
Confidence in the COVID-19 vaccine			
No	Ref	Ref	
Yes	6.45	2.13–19.5	0.001
Perceived susceptibility of contracting COVID-19	1.04	0.84–1.30	0.711
Perceived severity of COVID-19	0.75	0.54–1.05	0.090
Perceived benefits of the COVID-19 vaccine	1.85	1.27–2.70	0.001
Perceived barriers of the COVID-19 vaccine	0.97	0.80–1.19	0.793
Cues to action	1.48	1.03–2.11	0.029

## Data Availability

The raw data supporting the conclusions of this article will be made available by the authors if requested, without undue reservation.

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
