# Peer review of "Factors Affecting COVID-19 Vaccination Intentions among Patients Attending a Large HIV Treatment Clinic in Trinidad Using Constructs of the Health Belief Model"

_vaccines, 2022, doi:10.3390/vaccines11010004_

Round 1

Reviewer 1 Report

Factors Affecting COVID-19 Vaccination Intentions Among Patients Attending a Large HIV Treatment Clinic in Trinidad using Constructs of the Health Belief Model

Thank you very much for the invitation to review this paper. This paper has done several impressing works. However, this work needs improvements.  I have a couple of concerns regarding this study, and I recommend addressing these concerns before publishing;

Abstract:

The results need improvements, while no conclusion or recommendation is mentioned in this section. Please consider revising.

Introduction:

The introduction section needs to be developed properly. The whole section needs modifications and re-organization. The authors reported the relevant recent studies but need improvement; for example, studies mentioned [53-72] can be summarized more. The study objectives should be placed at the end of the introduction section.  

Minor:

Line 67 ; please correct COIVD-19  into COVID-19

Line 85: No need for the word Existing at the beginning of the sentence.

Methods:

The Material and Method part of the article should be written more clearly and understandably. For example, international readers need information about the study setting? The sampling procedure is not clear and mentioned in may sections of the methods. The inclusion and exclusion criteria written in a redundant way  example inclusion criteria “Patients 18 years and older”, no need to repeat   “Patients under 18 years of age in exclusion criteria”. Also, no information about how data is collected. I suggest you use STROBE guidelines for reporting the method: https://www.equator-network.org/reporting-guidelines/strobe/.    

I do not agree with the questions in the construct of Perceived Benefits of the COVID-19 vaccine [   I will be very sick if I take the COVID-19 vaccine and I am afraid of taking the COVID-19 vaccine]. Perceived benefits refer to a person's perception of the effectiveness of various actions available to reduce the threat of illness or disease. May be “eg, “COVID-19 vaccination can protect me from infection with SARS-CoV-2 or COVID-19.

Results:

The results section of the article should be written more clearly and understandably, for example it is written “High COVID-19 vaccination intention was less likely among females (52.2%; OR 0.49, 95% 230 CI 0.30-0.81) relative to males (68.9%). Which is hard for readers to understand. Odds ratio can be interpreted in more easy way. 

-        The calculated sample size was 262 participants, while the number of participants in the Tables 272, please explain?

-         Please provide the p value for all logistic regression coefficients for Table 3

-        Table 4 is hard to follow and need modifications.

Discussion:

Acceptable

Conclusion:

The conclusion revolves around the use of the HBM , this study can suggest some implications; however, the authors didn't develop the relevant implications. 

References:  

Please follow the references style of the Journal. In the text, reference numbers should be placed in square brackets [], rather than author names please consult https://www.mdpi.com/authors/references

Author Response

Response to Reviewer 1 Comments

Reviewer's comment: 

Abstract:

The results need improvements, while no conclusion or recommendation is mentioned in this section. Please consider revising.

Authors' response:

The abstract has been revised with results stated more understandably, with a conclusion and recommendation included.

Reviewer's comment:

Introduction:

The introduction section needs to be developed properly. The whole section needs modifications and re-organization. The authors reported the relevant recent studies but need improvement; for example, studies mentioned [53-72] can be summarized more. The study objectives should be placed at the end of the introduction section.  

Authors' response:

The introduction was re-organized with the relevant studies summarized and further literature relating the Health Belief Model in the context of COVID-19 vaccination has been included. The objective of the study was placed at the end of the introduction section in a single defined statement.

Reviewer's comment:

Methods:

The Material and Method part of the article should be written more clearly and understandably. For example, international readers need information about the study setting? The sampling procedure is not clear and mentioned in may sections of the methods. The inclusion and exclusion criteria written in a redundant way example inclusion criteria “Patients 18 years and older”, no need to repeat   “Patients under 18 years of age in exclusion criteria”. Also, no information about how data is collected. I suggest you use STROBE guidelines for reporting the method: https://www.equator-network.org/reporting-guidelines/strobe/.    

I do not agree with the questions in the construct of Perceived Benefits of the COVID-19 vaccine [   I will be very sick if I take the COVID-19 vaccine and I am afraid of taking the COVID-19 vaccine]. Perceived benefits refer to a person's perception of the effectiveness of various actions available to reduce the threat of illness or disease. May be “eg, “COVID-19 vaccination can protect me from infection with SARS-CoV-2 or COVID-19.

Authors' response:

The methods and materials section has been revised. Details of the study setting have been included for international readers, the sampling procedure has been summarized and repetition of the eligibility criteria has been addressed. The correct question relating to the perceived benefits of the COVID-19 vaccine have been included. These are the same questions presented in the results section.

Reviewer's comment:

Results:

The results section of the article should be written more clearly and understandably, for example it is written “High COVID-19 vaccination intention was less likely among females (52.2%; OR 0.49, 95% 230 CI 0.30-0.81) relative to males (68.9%). Which is hard for readers to understand. Odds ratio can be interpreted in more easy way. 

The calculated sample size was 262 participants, while the number of participants in the Tables 272, please explain.

Please provide the p value for all logistic regression coefficients for Table 3.

Table 4 is hard to follow and need modifications.

Authors' response:

The results section has been revised with vaccination intentions presented in terms of odds ratios. The justification for the use of n = 272 patients has been included in the methods section. Table 3 has now been presented as two tables instead, now Tables 3 and 4. P-values for the univariate logistic regression are presented in Table 4. The original Table 4 has now been presented as two tables, i.e. Tables 5 and 6, which have been modified for easier reading.

Reviewer's comment:

Conclusion:

The conclusion revolves around the use of the HBM, this study can suggest some implications; however, the authors didn't develop the relevant implications. 

Authors' response:

Implications surrounding the HBM in the context of the have been included in the conclusion.

Reviewer's comment:

References:  

Please follow the references style of the Journal. In the text, reference numbers should be placed in square brackets [], rather than author names please consult https://www.mdpi.com/authors/references

Authors' response:

Reference style has been updated to reflect the style of the Journal.

Reviewer 2 Report

First and foremost, I wish to congratulate the author for exploring such an important topic in a timely fashion. The study aims to understand COVID-19 vaccination intentions among HIV patients living in Trinidad under the HBM framework. Please find my comments below and respond to them sufficiently, as I believe answering these concerns could help the author further enhance her/his work, and in turn, the readers better appreciate the study.

1.     In the “Abstract” section, the authors mentioned, “Data were collected on the HBM constructs, in additional to socio-demographics and health factors”. Aside from the fact that this sentence contains a grammatical error (which can happen to anyone, but please sufficiently address this issue throughout the manuscript), could the authors please add an example to explain what they mean by “health factors” so that the readers would know how these factors differ from HBM constructs?

2.     Please add up-to-date literature regarding the use and application of the HBM framework in the COVID-19 context, as opposed to diseases like the seasonal flu.

3.     Could the authors please shed some light on how were the survey scales developed? For instance, were pre-tests used to validate the questionnaire before mass distribution?

4.     Could the authors please provide justifications for changing the “COVID vaccination intention” into a dichotomous variable? For instance, what are the pros and cons of this decision, and what factors make the authors believe the pros outweigh the cons?

Author Response

Responses to Reviewer 2 Comments:

Reviewer comment:

In the “Abstract” section, the authors mentioned, “Data were collected on the HBM constructs, in additional to socio-demographics and health factors”. Aside from the fact that this sentence contains a grammatical error (which can happen to anyone, but please sufficiently address this issue throughout the manuscript), could the authors please add an example to explain what they mean by “health factors” so that the readers would know how these factors differ from HBM constructs?

Authors' response:

The manuscript was reviewed for grammatical errors and corrected where necessary. In the method section, the distinction between health factors and HBM constructs has been made.

Reviewer's comment:

Please add up-to-date literature regarding the use and application of the HBM framework in the COVID-19 context, as opposed to diseases like the seasonal flu.

Authors' response:

In the introduction, recent and relevant literature on the use of the HBM in the COVID-19 context have been included.

Reviewer's comment:

Could the authors please shed some light on how were the survey scales developed? For instance, were pre-tests used to validate the questionnaire before mass distribution?

Authors' response:

The methods section was updated providing more information of survey instrument development and pre-testing.

Reviewer's comment:

Could the authors please provide justifications for changing the “COVID vaccination intention” into a dichotomous variable? For instance, what are the pros and cons of this decision, and what factors make the authors believe the pros outweigh the cons?

Authors' response:

Justification for the transformation and use of COVID-19 vaccination intention as a dichotomous variable has been provided.

Round 2

Reviewer 1 Report

ONLY minor issue the estimated midyear population is 1,367,558, not 1.39, for the year 2021, also please write the reference (14) in more detail.

Good Luck!